# A Structural Smoothing Framework For Robust Graph-Comparison

**Pinar Yanardag**
Department of Computer Science
Purdue University
West Lafayette, IN, 47906, USA
ypinar@purdue.edu

**S.V.N. Vishwanathan**
Department of Computer Science
University of California
Santa Cruz, CA, 95064, USA
vishy@ucsc.edu

## Abstract

In this paper, we propose a general smoothing framework for graph kernels by taking *structural similarity* into account, and apply it to derive smoothed variants of popular graph kernels. Our framework is inspired by state-of-the-art smoothing techniques used in natural language processing (NLP). However, unlike NLP applications that primarily deal with strings, we show how one can apply smoothing to a richer class of inter-dependent sub-structures that naturally arise in graphs. Moreover, we discuss extensions of the Pitman-Yor process that can be adapted to smooth structured objects, thereby leading to novel graph kernels. Our kernels are able to tackle the diagonal dominance problem while respecting the structural similarity between features. Experimental evaluation shows that not only our kernels achieve statistically significant improvements over the unsmoothed variants, but also outperform several other graph kernels in the literature. Our kernels are competitive in terms of runtime, and offer a viable option for practitioners.

## 1  Introduction

In many applications we are interested in computing similarities between structured objects such as graphs. For instance, one might aim to classify chemical compounds by predicting whether a compound is active in an anti-cancer screen or not. A kernel function which corresponds to a dot product in a reproducing kernel Hilbert space offers a flexible way to solve this problem [19]. R-convolution [10] is a framework for computing kernels between discrete objects where the key idea is to recursively decompose structured objects into sub-structures. Let $\langle \cdot, \cdot \rangle_{\mathcal{H}}$ denote a dot product in a reproducing kernel Hilbert space, $\mathcal{G}$ represent a graph and $\phi(\mathcal{G})$ represent a vector of sub-structure frequencies. The kernel between two graphs $\mathcal{G}$ and $\mathcal{G}'$ is computed by $\mathcal{K}(\mathcal{G}, \mathcal{G}') = \langle \phi(\mathcal{G}), \phi(\mathcal{G}') \rangle_{\mathcal{H}}$. Many existing graph kernels can be viewed as instances of R-convolution kernels. For instance, the graphlet kernel [22] decomposes a graph into graphlets, Weisfeiler-Lehman Subtree kernel (referred as Weisfeiler-Lehman for the rest of the paper) [23] decomposes a graph into subtrees, and the shortest-path kernel [1] decomposes a graph into shortest-paths. However, R-convolution based graph kernels suffer from a few drawbacks. First, the size of the feature space often grows exponentially. As size of the space grows, the probability that two graphs will contain similar sub-structures becomes very small. Therefore, a graph becomes similar to itself but not to any other graph in the training data. This is well known as the *diagonal dominance problem* [11] where the resulting kernel matrix is close to the identity matrix. Second, lower order sub-structures tend to be more numerous while a vast majority of the sub-structures occurs rarely. In other words, a few sub-structures dominate the distribution. This exhibits a strong power-law behavior and results in underestimation of the true distribution. Third, the sub-structures used to define a graph kernel are often related to each other. However, an R-convolution kernel only respects exact matchings. This problem is particu-

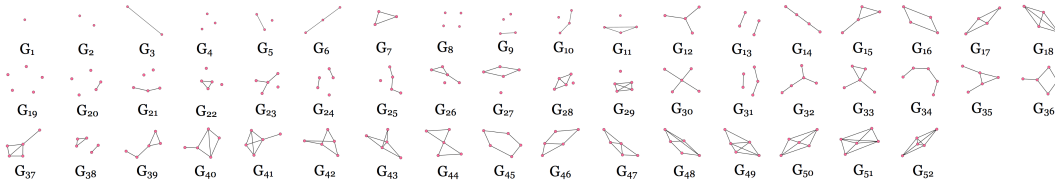

Figure 1: Graphlets of size $k \leq 5$.

larly important when noise is present in the training data and considering partial similarity between sub-structures might alleviate the noise problem.

**Our solution:** In this paper, we propose to tackle the above problems by using a general framework to *smooth* graph kernels that are defined using a frequency vector of decomposed structures. We use *structure* information by encoding relationships between lower and higher order sub-structures in order to derive our method. The remainder of this paper is structured as follows. In Section 2, we review three families of graph kernels for which our smoothing is applicable. In Section 3, we review smoothing methods for multinomial distributions. In Section 4, we introduce a framework for smoothing structured objects. In Section 5, we propose a Bayesian variant of our model that is extended from the Hierarchical Pitman-Yor process [25]. In Section 6, we discuss related work. In Section 7, we compare smoothed graph kernels to their unsmoothed variants as well as to other state-of-the-art graph kernels. We report results on classification accuracy on several benchmark datasets as well as their noisy-variants. Section 8 concludes the paper.

## 2 Graph kernels

Existing graphs kernels based on R-convolution can be categorized into three major families: graph kernels based on limited-sized subgraphs [*e.g.* 22], graph kernels based on subtree patterns [*e.g.* 18, 21], and graph kernels based on walks [*e.g.* 27] or paths [*e.g.* 1].

**Graph kernels based on subgraphs:** A *graphlet* $G$ [17] is non-isomorphic sub-graph of size-$k$, (see Figure 1). Given two graphs $\mathcal{G}$ and $\mathcal{G}'$, the kernel [22] is defined as $\mathcal{K}_{GK}(\mathcal{G}, \mathcal{G}') = \left\langle \mathbf{f}^{\mathcal{G}}, \mathbf{f}^{\mathcal{G}'} \right\rangle$ where $\mathbf{f}^{\mathcal{G}}$ and $\mathbf{f}^{\mathcal{G}'}$ are vectors of normalized counts of graphlets, that is, the $i$-th component of $\mathbf{f}^{\mathcal{G}}$ (resp. $\mathbf{f}^{\mathcal{G}'}$) denotes the frequency of graphlet $G_i$ occurring as a sub-graph of $\mathcal{G}$ (resp. $\mathcal{G}'$).

**Graph kernels based on subtree patterns:** Weisfeiler-Lehman [21] is a popular instance of graph kernels that decompose a graph into its subtree patterns. It simply iterates over each vertex in a graph, and compresses the label of the vertex and labels of its neighbors into a multiset label. The vertex is then relabeled with the compressed label to be used for the next iteration. Algorithm concludes after running for $h$ iterations, and the compressed labels are used for constructing a frequency vector for each graph. Formally, given $\mathcal{G}$ and $\mathcal{G}'$, this kernel is defined as $\mathcal{K}_{WL}(\mathcal{G}, \mathcal{G}') = \left\langle \mathbf{l}^{\mathcal{G}}, \mathbf{l}^{\mathcal{G}'} \right\rangle$ where $\mathbf{l}^{\mathcal{G}}$ contains the frequency of each compressed label occurring in $h$ iterations.

**Graph kernels based on walks or paths:** Shortest-path graph kernel [1] is a popular instance of this family. This kernel simply compares the sorted endpoints and the length of shortest-paths that are common between two graphs. Formally, let $\mathbb{P}_{\mathcal{G}}$ represent the set of all shortest-paths in graph $\mathcal{G}$, and $p_i \in \mathbb{P}_{\mathcal{G}}$ denote a triplet $(l_s, l_e, n_k)$ where $n_k$ is the length of the path and $l_s$ and $l_e$ are the labels of the source and sink vertices, respectively. The kernel between graphs $\mathcal{G}$ and $\mathcal{G}'$ is defined as $\mathcal{K}_{SP}(\mathcal{G}, \mathcal{G}') = \left\langle \mathbf{p}^{\mathcal{G}}, \mathbf{p}^{\mathcal{G}'} \right\rangle$ where $i$-th component of $\mathbf{p}^{\mathcal{G}}$ contains the frequency of $i$-th triplet occurring in graph $\mathcal{G}$ (resp. $\mathbf{p}^{\mathcal{G}'}$).

## 3 Smoothing multinomial distributions

In this section, we briefly review smoothing techniques for multinomial distributions. Let $e_1, e_2, \ldots, e_m$ represent a sequence of $n$ discrete events drawn from a ground set $\mathcal{A} = \{1, 2, \ldots, V\}$.

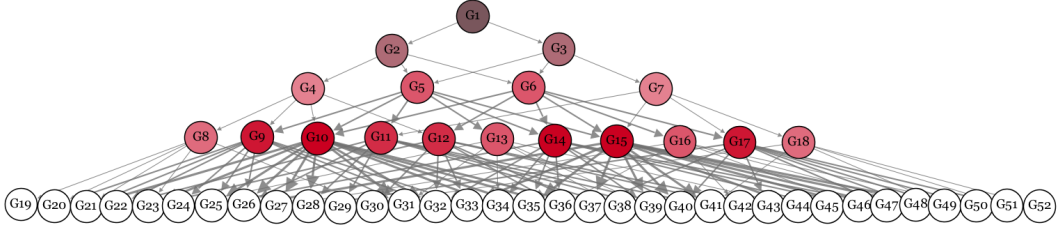

Figure 2: Topologically sorted graphlet DAG for $k \leq 5$ where nodes are colored based on degree.

Suppose, we would like to estimate the probability $P\left(e_i = a\right)$ for some $a \in \mathcal{A}$. It is well known that the Maximum Likelihood Estimate (MLE) can be computed as $P_{MLE}\left(e_i = a\right) = \frac{c_a}{m}$ where $c_a$ denotes the number of times the event $a$ appears in the observed sequence and $m = \sum_j c_j$ denotes the total number of observed events. However, MLE of the multinomial distribution is *spiky* since it assigns zero probability to the events that did not occur in the observed sequence. In other words, an event with low probability is often estimated to have zero probability mass. The general idea behind smoothing is to adjust the MLE of the probabilities by pushing the high probabilities downwards and pushing low or zero probabilities upwards in order to produce a more accurate distribution on the events [30]. Interpolated smoothing methods offer a flexible solution between the higher-order maximum likelihood model and lower-order smoothed model (or so-called, *fallback* model). The way the fallback model is designed is the key to define a new smoothing method[1]. Absolute discounting [15] and Interpolated Kneser-Ney [12] are two popular instances of interpolated smoothing methods:

$$P_A\left(e_i = a\right) = \frac{\max\left\{c_a - d, 0\right\}}{m} + \frac{m_d \times d}{m} P'_A\left(e_i = a\right). \quad (1)$$

Here, $d > 0$ is a discount factor, $m_d := \left|\{a : c_a > d\}\right|$ is the number of events whose counts are larger than $d$, while $P'_A$ is the fallback distribution. Absolute discounting defines the fallback distribution as the smoothed version of the lower-order MLE while Kneser-Ney uses an unusual estimate of the fallback distribution by using number of different contexts that the event follows in the lower order model.

## 4 Smoothing structured objects

In this section, we first propose a new interpolated smoothing framework that is applicable to a richer set of objects such as graphs by using a Directed Acyclic Graph (DAG). We then discuss how to design such DAGs for various graph kernels.

### 4.1 Structural smoothing

The key to designing a new smoothing method is to define a fallback distribution, which not only incorporates domain knowledge but is also easy to estimate recursively. Suppose, we have access to a weighted DAG where every node at the $k$-th level represents an event from the ground set $\mathcal{A}$. Moreover let $w_{ij}$ denote the weight of the edge connecting event $i$ to event $j$, and $\mathcal{P}_a$ (resp. $\mathcal{C}_a$) denote the parents (resp. children) of event $a \in \mathcal{A}$ in the DAG. We define our structural smoothing for events at level $k$ as follows:

$$P_{SS}^k\left(e_i = a\right) = \frac{\max\left\{c_a - d, 0\right\}}{m} + \frac{m_d \times d}{m} \sum_{j \in \mathcal{P}_a} P_{SS}^{k-1}\left(j\right) \frac{w_{ja}}{\sum_{a' \in \mathcal{C}_j} w_{ja'}}. \quad (2)$$

The way to understand the above equation is as follows: we subtract a fixed discounting factor $d$ from every observed event which accumulates to a total mass of $m_d \times d$. Each event $a$ receives some portion of this accumulated probability mass from its parents. The proportion of the mass that a parent $j$ at level $k-1$ transmits to a given child $a$ depends on the weight $w_{ja}$ between the parent and the child (normalized by the sum of the weights of the edges from $j$ to all its children), and the probability mass $P_{SS}^{k-1}\left(j\right)$ that is assigned to node $j$. In other words, the portion a child event $a$ is able to obtain from the total discounted mass depends on how authoritative its parents are, and how strong the relationship between the child and its parents.

## 4.2 Designing the DAG

In order to construct a DAG for smoothing structured objects, we first construct a vocabulary $V$ that denotes the set of all unique sub-structures that are going to be *smoothed*. Each item in the vocabulary $V$ corresponds to a node in the DAG. $V$ can be generated statically or dynamically based on the type of sub-structure the graph kernel exploits. For instance, it requires a one-time $O(2^k)$ effort to generate the vocabulary of size $\leq k$ graphlets for graphlet kernel. However, one needs to build the vocabulary dynamically in Weisfeiler-Lehman and Shortest-Path kernels since the sub-structures depend on the node labels obtained from the datasets. After constructing the vocabulary $V$, the parent/child relationship between sub-structures needs to be obtained. Given a sub-structure $s$ of size $k$, we apply a *transformation* to find all possible sub-structures of size $k-1$ that $s$ can be reduced into. Each sub-structure $s'$ that is obtained by this transformation is assigned as a *parent* of $s$. After obtaining the parent/child relationship between sub-structures, the DAG is constructed by drawing a directed edge from each parent to its children nodes. Since all descendants of a given sub-structure at depth $k-1$ are at depth $k$, this results in a topological ordering of the vertices, and hence the resulting graph is indeed a DAG. Next, we discuss how to construct such DAGs for different graph kernels.

**Graphlet Kernel:** We construct the vocabulary $V$ for graphlet kernel by enumerating all canonical graphlets of size up to $k$[2]. Each canonically-labeled graphlet is a node in the DAG. We then apply a transformation to infer the parent/child relationship between graphlets as follows: we place a directed edge from graphlet $G$ to $G'$ if, and only if, $G$ can be obtained from $G'$ by deleting a node. In other words, all edges from a graphlet $G$ of size $k-1$ point to a graphlet $G'$ of size $k$. In order to assign weights to the edges, given a graphlet pair $G$ and $G'$, we count the number of times $G$ can be obtained from $G'$ by deleting a node (call this number $n_{GG'}$). Recall that $G$ is of size $k-1$ and $G'$ is of size $k$, and therefore $n_{GG'}$ can at most be $k$. Let $\mathcal{C}_G$ denote the set of children of node $G$ in the DAG, and $n_G := \sum_{\bar{G} \in \mathcal{C}_G} n_{G\bar{G}}$. Then we define the weight $w_{GG'}$ of the edge connecting $G$ and $G'$ as $n_{GG'}/n_G$. The idea here is that the weight encodes the proportion of different ways of extending $G$ which results in the graphlet $G'$. For instance, let us consider $G_{15}$ and its parents $G_5, G_6, G_7$ (see Figure 2 for the DAG of graphlets with size $k \leq 5$). Even if graphlet $G_{15}$ is not observed in the training data, it still gets a probability mass proportional to the edge weight from its parents in order to overcome the sparsity problem of unseen data.

**Weisfeiler-Lehman Kernel:** The Weisfeiler-Lehman kernel performs an *exact matching* between the compressed multiset labels. For instance, given two labels ABCD**E** and ABCD**F**, it simply assigns zero value for their similarity even though two labels have a partial similarity. In order to smooth Weisfeiler-Lehman kernel, we first run the original algorithm and obtain the multiset representation of each graph in the dataset. We then apply a transformation to infer the parent/child relationship between compressed labels as follows: in each iteration of Weisfeiler-Lehman algorithm, and for each multiset label of size $k$ in the vocabulary, we generate its *power set* by computing all subsets of size $k-1$ while keeping the root node fixed. For instance, the parents of a multiset label ABCDE are {ABCD, ABCE, ABDE, ACDE}. Then, we simply construct the DAG by drawing a directed edge from parent labels to children. Notice that considering only the set of labels generated from the Weisfeiler-Lehman kernel is not sufficient enough for constructing a valid DAG. For instance, it might be the case that none of the possible parents of a given label exists in the vocabulary simply due to the sparsity problem (*e.g.*out of all possible parents of ABCDE, we might only observe ABCE in the training data). Thus, restricting ourselves to the original vocabulary leaves such labels orphaned in the DAG. Therefore, we consider so-called *pseudo parents* as a part of the vocabulary when constructing the DAG. Since the sub-structures in this kernel are data-dependent, we use a uniform weight between a parent and its children.

**Shortest-Path Kernel:** Similar to other graph kernels discussed above, shortest-path graph kernel does not take partial similarities into account. For instance, given two shortest-paths ABCD**E** and ABCD**F** (compressed as AE5 and AF5, respectively), it assigns zero for their similarity since their sink labels are different. However, one can notice that shortest-path sub-structures exhibit a strong dependency relationship. For instance, given a shortest-path $p_{ij} = \{ABCDE\}$ of size $k$, one can derive the shortest-paths {ABCD, ABC, AB} of size $< k$ as a result of the *optimal sub-structure* property, that is, one can show that all sub-paths of a shortest-path are also shortest-paths with

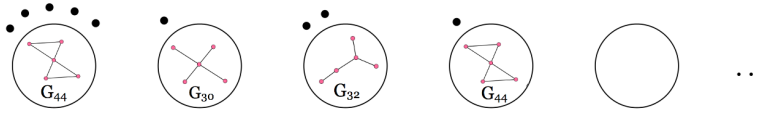

Figure 3: An illustration of table assignment, adapted from [9]. In this example, labels at the tables are given by $(l_1, \ldots, l_4) = (G_{44}, G_{30}, G_{32}, G_{44})$. Black dots indicate the number of occurrences of each label in 10 draws from the Pitman-Yor process.

the same source node [6]. In order to smooth shortest-path kernel, we first build the vocabulary by computing all shortest-paths for each graph. Let $p_{ij}$ be a shortest-path of size $k$ and $p_{ij'}$ be a shortest-path of size $k-1$ that is obtained by removing the sink node of $p_{ij}$. Let $l_{ij}$ be the compressed form of $p_{ij}$ that represents the sorted labels of its endpoints $i$ and $j$ concatenated to its length (resp. $l_{ij'}$). Then, in order to build the DAG, we draw a directed edge from $l_{ij'}$ of depth $k-1$ to $l_{ij}$ of depth $k$ if and only if $p_{ij'}$ is a sub-path of $p_{ij}$. In other words, all ascendants of $l_{ij}$ consist of the compressed labels obtained from sub-paths of $p_{ij}$ of size $< k$. Similar to Weisfeiler-Lehman kernel, we assign a uniform weight between parents and children.

## 5 Pitman-Yor Smoothing

Pitman-Yor processes are known to produce power-law distributions [8]. A novel interpretation of interpolated Kneser-Ney is proposed by [25] as approximate inference in a hierarchical Bayesian model consisting of Pitman-Yor process [16]. By following a similar spirit, we extend our model to adapt Pitman-Yor process as an alternate smoothing framework. A Pitman-Yor process $P$ on a ground set $\mathcal{G}_{k+1}$ of size-$(k+1)$ graphlets is defined via $P_{k+1} \sim PY(d_{k+1}, \theta_{k+1}, P_k)$ where $d_{k+1}$ is a discount parameter, $0 \leq d_{k+1} < 1$, $\theta > -d_{k+1}$ is a strength parameter, and $P_k$ is a base distribution. The most intuitive way to understand draws from the Pitman-Yor process is via the Chinese restaurant process (see Figure 3). Consider a restaurant with an infinite number of tables

---

**Algorithm 1** Insert a Customer

**Input:** $d_{k+1}, \theta_{k+1}, P_k$

  $t \leftarrow 0$  // Occupied tables
  $c \leftarrow ()$  // Counts of customers
  $l \leftarrow ()$  // Labels of tables
  **if** $t = 0$ **then**
    $t \leftarrow 1$
    append 1 to $c$
    draw graphlet $G_i \sim P_k$  // Insert customer in parent
    draw $G_j \sim w_{ij}$
    append $G_j$ to $l$
    **return** $G_j$
  **else**
    with probability $\propto \max(0, c_j - d)$
    $c_j \leftarrow c_j + 1$
    **return** $l_j$
    with probability proportional to $\theta + dt$
    $t \leftarrow t + 1$
    append 1 to $c$
    draw graphlet $G_i \sim P_k$  // Insert customer in parent
    draw $G_j \sim w_{ij}$
    append $G_j$ to $l$
    **return** $G_j$
  **end if**

---

where customers enter the restaurant one by one. The first customer sits at the first table, and a graphlet is assigned to it by drawing a sample from the base distribution since this table is occupied for the first time. The label of the first table is the first graphlet drawn from the Pitman-Yor process.

Subsequent customers when they enter the restaurant decide to sit at an already occupied table with probability proportional to $c_i - d_{k+1}$, where $c_i$ represents the number of customers already sitting at table $i$. If they sit at an already occupied table, then the label of that table denotes the next graphlet drawn from the Pitman-Yor process. On the other hand, with probability $\theta_{k+1} + d_{k+1}t$, where $t$ is the current number of occupied tables, a new customer might decide to occupy a new table. In this case, the base distribution is invoked to label this table with a graphlet. Intuitively the reason this process generates power-law behavior is because popular graphlets which are served on tables with a large number of customers have a higher probability of attracting new customers and hence being generated again, similar to a rich gets richer phenomenon. In a hierarchical Pitman-Yor process, the base distribution $P_k$ is recursively defined via a Pitman-Yor process $P_k \sim PY(d_k, \theta_k, P_{k-1})$. In order to label a table, we need a draw from $P_k$, which is obtained by inserting a customer into the corresponding restaurant. However, adopting the traditional hierarchical Pitman-Yor process is not straightforward in our case since the *size* of the context differs between levels of hierarchy, that is, a *child* restaurant in the hierarchy can have more than one *parent* restaurant to request a label from. In other words, $P_{k+1}$ is defined over $\mathcal{G}_{k+1}$ of size $n_{k+1}$ while $P_k$ is defined over $\mathcal{G}_k$ of size $n_k \leq n_{k+1}$. Therefore, one needs a *transformation function* to transform base distributions of different sizes. We incorporate edge weights between parent and child restaurants by using the same weighting scheme in Section 4.2. This changes the Chinese Restaurant process as follows: When we need to label a table, we will first draw a size-$k$ graphlet $G_i \sim P_k$ by inserting a customer into the corresponding restaurant. Given $G_i$, we will draw a size-$(k+1)$ graphlet $G_j$ proportional to $w_{ij}$, where $w_{ij}$ is obtained from the DAG. See Algorithm 1 for pseudo code of inserting a customer. Deletion of a customer is handled similarly (see Algorithm 2).

---

**Algorithm 2** Delete a Customer

---

**Input:** $d, \theta, P_0, C, L, t$
  with probability $\propto c_l$
  $c_l \leftarrow c_l - 1$
  $G_j \leftarrow l_j$
  **if** $c_l = 0$ **then**
    $P_k \propto 1/w_{ij}$
    delete $c_l$ from $c$
    delete $l_j$ from $l$
    $t \leftarrow t - 1$
  **end if**
  **return** $G$

---

# 6 Related work

A survey of most popular graph kernel methods is already given in previous sections. Several methods proposed in smoothing structured objects [4], [20]. Our framework is similar to dependency tree kernels [4] since both methods are using the notion of smoothing for structured objects. However, our method is interested in the problem of smoothing the count of structured objects. Thus, while smoothing is achieved by using a DAG, we discard the DAG once the counts are smoothed. Another related work to ours is propagation kernels [14] that define graph features as counts of similar node-label distributions on the respective graphs by using Locality Sensitive Hashing (LSH). Our framework not only considers node label distributions, but also explicitly incorporates structural similarity via the DAG. Another similar work to ours is recently proposed framework by [29] which learns the co-occurrence relationship between sub-structures by using neural language models. However, their framework do not respect the structural similarity between sub-structures, which is an important property to consider especially in the presence of noise in edges or labels.

# 7 Experiments

The aim of our experiments is threefold. First, we want to show that smoothing graph kernels significantly improves the classification accuracy. Second, we want to show that the smoothed kernels are comparable to or outperform state-of-the-art graph kernels in terms of classification

Table 1: Comparison of classification accuracy ($\pm$ standard deviation) of shortest-path (SP), Weisfeiler-Lehman (WL), graphlet (GK) kernels with their smoothed variants. Smoothed variants with statistically significant improvements over the base kernels are shown in bold as measured by a $t$-test with a $p$ value of $\leq 0.05$. Ramon & Gärtner (Ram & Gär), $p$-random walk and random walk kernels are included for additional comparison where $> 72$H indicates the computation did not finish in 72 hours. Runtime for constructing the DAG and smoothing (SMTH) the counts are also reported where " indicates seconds and ' indicates minutes.

| DATASET | MUTAG | | PTC | | ENZYMES | | PROTEINS | | NCI1 | | NCI109 | |
|---|---|---|---|---|---|---|---|---|---|---|---|---|
| SP | $85.22_{\pm2.43}$ | | $58.24_{\pm2.44}$ | | $40.10_{\pm1.50}$ | | $75.07_{\pm0.54}$ | | $73.00_{\pm0.24}$ | | $73.00_{\pm0.21}$ | |
| SMOOTHED SP | $\mathbf{87.94}_{\pm2.58}$ | | $\mathbf{60.82}_{\pm1.84}$ | | $\mathbf{42.27}_{\pm1.07}$ | | $\mathbf{75.85}_{\pm0.28}$ | | $\mathbf{73.26}_{\pm0.24}$ | | $73.01_{\pm0.31}$ | |
| WL | $82.22_{\pm1.87}$ | | $60.41_{\pm1.93}$ | | $53.88_{\pm0.95}$ | | $74.49_{\pm0.49}$ | | $84.13_{\pm0.22}$ | | $83.83_{\pm0.31}$ | |
| SMOOTHED WL | $\mathbf{87.44}_{\pm1.95}$ | | $60.47_{\pm2.39}$ | | $\mathbf{55.30}_{\pm0.65}$ | | $\mathbf{75.53}_{\pm0.50}$ | | $\mathbf{84.66}_{\pm0.18}$ | | $\mathbf{84.72}_{\pm0.21}$ | |
| GK | $81.33_{\pm1.02}$ | | $55.56_{\pm1.46}$ | | $27.32_{\pm0.96}$ | | $69.69_{\pm0.46}$ | | $62.46_{\pm0.19}$ | | $62.33_{\pm0.14}$ | |
| SMOOTHED GK | $\mathbf{83.17}_{\pm0.64}$ | | $\mathbf{58.44}_{\pm1.00}$ | | $\mathbf{30.90}_{\pm1.51}$ | | $69.83_{\pm0.46}$ | | $62.48_{\pm0.15}$ | | $\mathbf{62.48}_{\pm0.11}$ | |
| PYP GK | $\mathbf{83.11}_{\pm1.23}$ | | $57.44_{\pm1.44}$ | | $29.63_{\pm1.30}$ | | $70.00_{\pm0.80}$ | | $62.50_{\pm0.20}$ | | $\mathbf{62.68}_{\pm0.18}$ | |
| RAM & GÄR | $84.88_{\pm1.86}$ | | $58.47_{\pm0.90}$ | | $16.96_{\pm1.46}$ | | $70.73_{\pm0.35}$ | | $56.61_{\pm0.53}$ | | $54.62_{\pm0.23}$ | |
| P-RANDOMWALK | $80.05_{\pm1.64}$ | | $59.38_{\pm1.66}$ | | $30.01_{\pm1.00}$ | | $71.16_{\pm0.35}$ | | $>72$H | | $>72$H | |
| RANDOM WALK | $83.72_{\pm1.50}$ | | $57.85_{\pm1.30}$ | | $24.16_{\pm1.64}$ | | $74.22_{\pm0.42}$ | | $>72$H | | $>72$H | |
| DAG/SMTH (GK) | 6" | 1" | 6" | 1" | 6" | 1" | 6" | 1" | 6" | 3" | 6" | 3" |
| DAG/SMTH (SP) | 3" | 1" | 19" | 1" | 45" | 1" | 9' | 1" | 9' | 17" | 10' | 16" |
| DAG/SMTH (WL) | 1" | 2" | 1" | 17" | 10" | 12' | 7' | 70' | 2" | 21' | 2" | 21' |
| DAG/SMTH (PYP) | 6" | 5" | 6" | 12" | 6" | 21" | 6" | 1' | 6" | 8' | 6" | 8' |

accuracy, while remaining competitive in terms of computational requirements. Third, we want to show that our methods outperform base kernels when edge or label noise is presence.

**Datasets** We used the following benchmark datasets used in graph kernels: MUTAG, PTC, EN-ZYMES, PROTEINS, NCI1 and NCI109. MUTAG is a dataset of 188 mutagenic aromatic and heteroaromatic nitro compounds [5] with 7 discrete labels. PTC [26] is a dataset of 344 chemical compounds has 19 discrete labels. ENZYMES is a dataset of 600 protein tertiary structures obtained from [2], and has 3 discrete labels. PROTEINS is a dataset of 1113 graphs obtained from [2] having 3 discrete labels. NCI1 and NCI109 [28] are two balanced datasets of chemical compounds having size 4110 and 4127 with 37 and 38 labels, respectively.

**Experimental setup** We compare our framework against representative instances of major families of graph kernels in the literature. In addition to the base kernels, we also compare our smoothed kernels with the random walk kernel [7], the Ramon-Gärtner subtree [18], and $p$-step random walk kernel [24]. The Random Walk, $p$-step Random Walk and Ramon-Gärtner are written in Matlab and obtained from [22]. All other kernels were coded in Python except Pitman-Yor smoothing which is coded in C++[3]. We used a parallel implementation for smoothing the counts of Weisfeiler-Lehman kernel for efficiency. All kernels are normalized to have a unit length in the feature space. Moreover, we use 10-fold cross validation with a binary $C$-Support Vector Machine (SVM) where the $C$ value for each fold is independently tuned using training data from that fold. In order to exclude random effects of the fold assignments, this experiment is repeated 10 times and average prediction accuracy of 10 experiments with their standard deviations are reported[4].

## 7.1 Results

In our first experiment, we compare the base kernels with their smoothed variants. As can be seen from Table 1, smoothing improves the classification accuracy of *every base kernel* on *every dataset* with majority of the improvements being statistically significant with $p \leq 0.05$. We observe that even though smoothing improves the accuracy of graphlet kernels on PROTEINS and NCI1, the improvements are not statistically significant. We believe this is due to the fact that these datasets are not sensitive to structural noise as much as the other datasets, thus considering the partial similarities

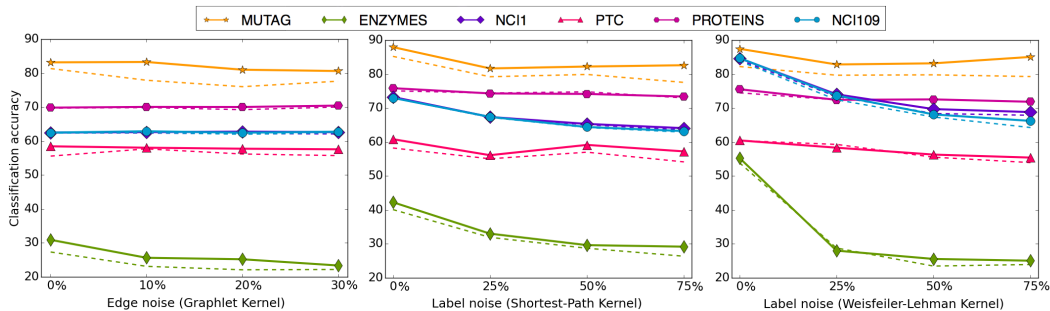

Figure 4: Classification accuracy vs. noise for base graph kernels (dashed lines) and their smoothed variants (non-dashed lines).

do not improve the results significantly. Moreover, PYP smoothed graphlet kernels achieve statistically significant improvements in most of the datasets, however they are outperformed by smoothed graphlet kernels introduced in Section 3.

In our second experiment, we picked the best smoothed kernel in terms of classification accuracy for each dataset, and compared it against the performance of state-of-the-art graph kernels (see Table 1). Smoothed kernels outperform other methods on all datasets, and the results are statistically significant on every dataset except PTC.

In our third experiment, we investigated the runtime behavior of our framework with two major costs. First, one has to compute a DAG by using the original feature vectors. Next, the constructed DAG need to be used to compute *smoothed* representations of the feature vectors. Table 1 shows the total wallclock runtime taken by *all graphs* for constructing the DAG, and smoothing the counts for each dataset. As can be seen from the runtimes, our framework adds a constant factor to the original runtime for most of the datasets. While the DAG creation in Weisfeiler-Lehman kernel also adds a negligible overhead, the cost of smoothing becomes significant if the vocabulary size gets prohibitively large due to the exponential growing nature of the kernel *w.r.t.* to subtree parameter $h$.

Finally, in our fourth experiment, we test the performance of graph kernels when *edge* or *label* noise is present. For edge noise, we randomly removed and added $\{10\%, 20\%, 30\%\}$ of the edges in each graph. For label noise, we randomly flipped $\{25\%, 50\%, 75\%\}$ of the node labels in each graph where random labels are selected proportionally to the original label-distribution of the graph. Figure 4 shows the performance of smoothed graph kernels under noise. As can be seen from the figure, smoothed kernels are able to outperform their base variants when noise is present. An interesting observation is that even though a significant amount of edge noise is added to PROTEINS and NCI datasets, the performance of base kernels do not change drastically. This further supports our observation that these datasets are not sensitive to structural noise as much as the other datasets.

# 8   Conclusion and Future Work

We presented a novel framework for smoothing graph kernels inspired by smoothing techniques from natural language processing and applied our method to state-of-the-art graph kernels. Our framework is rather general, and lends itself to many extensions. For instance, by defining domain-specific parent-child relationships, one can construct different DAGs with different weighting schemes. Another interesting extension of our smoothing framework would be to apply it to graphs with continuous labels. Moreover, even though we restricted ourselves to graph kernels in this paper, our framework is applicable to any R-convolution kernel that uses a frequency-vector based representation, such as *string kernels*.

# 9   Acknowledgments

We thank to Hyokun Yun for his tremendous help in implementing Pitman-Yor Processes. We also thank to anonymous NIPS reviewers for their constructive comments, and Jiasen Yang, Joon Hee Choi, Amani Abu Jabal and Parameswaran Raman for reviewing early drafts of the paper. This work is supported by the National Science Foundation under grant No. #1219015.

## Footnotes

[1]See Table 2 in [3] for summarization of various smoothing algorithms using this general framework.

[2]We used Nauty [13] to obtain canonically-labeled isomorphic representations of graphlets.

[3]We modified the open source implementation of PYP: https://github.com/redpony/cpyp.

[4]Implementations of original and smoothed versions of the kernels, datasets and detailed discussion of parameter selection procedure with the list of parameters used in our experiments can be accessed from http://web.ics.purdue.edu/~ypinar/nips.

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
