[Supplementary Material · nips2015_mainpaper.pdf]

# A Structural Smoothing Framework For Robust Graph-Comparison

**Pinar Yanardag**
Department of Computer Science
Purdue University
West Lafayette, IN, 47906, USA
ypinar@purdue.edu

**S.V.N. Vishwanathan**
Department of Computer Science
University of California
Santa Cruz, CA, 95064, USA
vishy@ucsc.edu

## Abstract

In this paper, we propose a general smoothing framework for graph kernels by taking *structural similarity* into account, and apply it to derive smoothed variants of popular graph kernels. Our framework is inspired by state-of-the-art smoothing techniques used in natural language processing (NLP). However, unlike NLP applications which primarily deal with strings, we show how one can apply smoothing to a richer class of inter-dependent sub-structures that naturally arise in graphs. Moreover, we discuss extensions of the Pitman-Yor process that can be adapted to smooth structured objects thereby leading to novel graph kernels. Our kernels are able to tackle the diagonal dominance problem, while respecting the structural similarity between sub-structures, especially under the presence of edge or label noise. Experimental evaluation shows that not only our kernels outperform the unsmoothed variants, but also achieve statistically significant improvements in classification accuracy over several other graph kernels that have been recently proposed in literature. Our kernels are competitive in terms of runtime, and offer a viable option for practitioners.

## 1   Introduction

In many applications we are interested in computing similarities between structured objects such as graphs. For instance, one might aim to classify chemical compounds by predicting whether a compound is active in an anti-cancer screen or not. A kernel function which corresponds to a dot product in a reproducing kernel Hilbert space offers a flexible way to solve this problem [18]. R-convolution [10] is a framework for computing kernels between discrete objects where the key idea is to recursively decompose structured objects into sub-structures. Let $\langle \cdot, \cdot \rangle_{\mathcal{H}}$ denote a dot product in a reproducing kernel Hilbert space, $\mathcal{G}$ represent a graph and $\phi(\mathcal{G})$ denote a vector of sub-structure frequencies. The kernel between two graphs $\mathcal{G}$ and $\mathcal{G}'$ is computed by $k(\mathcal{G}, \mathcal{G}') = \langle \phi(\mathcal{G}), \phi(\mathcal{G}') \rangle_{\mathcal{H}}$. Many existing graph kernels can be viewed as instance of R-convolution. For instance, the graphlet kernel [21] decomposes a graph into graphlets, Weisfeiler-Lehman Subtree kernel (referred as Weisfeiler-Lehman for the rest of the paper) [22] decomposes a graph into subtrees, and the shortest-path kernel [1] decomposes a graph into shortest-paths. However, R-convolution based graph kernels suffer from a few drawbacks. First, the size of the feature space often grows exponentially. As size of the space grows, the probability that two graphs will contain similar sub-structures becomes very small. Therefore, a graph becomes similar to itself but not to any other graph in the training data. This is well known as the *diagonal dominance problem*, and the resulting kernel matrix is close to the identity matrix. In other words, the graphs are orthogonal to each other in the feature space. Second, lower order sub-structures tend to be more numerous while a vast majority of the sub-structures occur very rarely. In other words, a few sub-structures dominate the distribution which exhibits a strong power-law behavior and results in underestimation of the true distribution.

Third, the sub-structures used to define a graph kernel are often related to each other. However, an R-convolution kernel only respects exact matchings. This problem is particularly important when noise is present in the data since considering partial similarity between sub-structures might alleviate the noise problem.

Figure 1: Graphlets of size $k \leq 5$.

**Our solution:** In this paper, we propose to tackle the above problems by using a general framework to *smooth* graph kernels that are defined using a frequency vector of decomposed structures. We use *structure* information by encoding relationships between lower order and higher order substructures in order to derive our method. Consequently, our smoothing algorithm not only respects the dependency between sub-structures but also tackles the diagonal dominance problem by distributing the probability mass across features.

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

**Datasets**   We experimented with the following standard benchmark datasets used in graph kernels: MUTAG, PTC, ENZYMES, PROTEINS, NCI1 and NCI109. MUTAG is a dataset of 188 mutagenic aromatic and heteroaromatic nitro compounds [5] with 7 discrete labels. PTC [25] is a dataset of 344 chemical compounds has 19 discrete labels. NCI1 and NCI109 [27] are two balanced datasets of chemical compounds having size 4110 and 4127 with 37 and 38 labels, respectively. ENZYMES is a dataset of 600 protein tertiary structures obtained from [2], and has 3 discrete labels. PROTEINS is a dataset of 1113 graphs obtained from [2] where nodes are secondary structure elements (SSEs) and there is an edge between two nodes if they are neighbors in the amino acid sequence or in 3D space, having 3 discrete labels. See Appendix **??** for more information on the datasets.

**Experimental setup**   We compare our framework against representative instances of major families of graph kernels in the literature. Other than base kernels of our framework; the Weisfeiler-Lehman kernel [21], the graphlet kernel [21], and the shortest-path kernel [1], we also compare our smoothed kernels with the random walk kernel [7], the subtree kernel [17], and $p$-step random walk kernel [23]. The Random Walk, $p$-step Random Walk and Ramon-Gärtner kernels are written in Matlab and were obtained from the authors of [21]. All other kernels were coded in Python[3]. We used a parallel implementation for smoothing the counts of Weisfeiler-Lehman kernel for efficiency purposes. In order to ensure a fair comparison, all experiments are performed on the same hardware. All kernels are normalized to have a unit length in the feature space. Moreover, we use 10-fold cross validation with a binary $C$-Support Vector Machine (SVM) to test classification performance. The $C$ value for each fold is independently tuned using training data from that fold. In order to exclude random effects of the fold assignments, this experiment is repeated 10 times and average prediction accuracy of 10 experiments with their standard deviations are reported. See Appendix **??** for a detailed discussion of parameter selection procedure for each algorithm and the parameters used in our experiments.

Figure 4: Classification accuracy vs. noise for base graph kernels (dashed lines) and their smoothed variants (non-dashed lines).

## 7.1 Results

In our first experiment, we compare the graphlet kernel, the Weisfeiler-Lehman kernel, and the shortest-path kernel with their smoothed variants. The results are in Table 1 where smoothed variants that are statistically significant over the base kernels are shown in bold as measured by a $t$-test with a $p$ value of $\leq 0.05$. As can be seen on *every dataset*, smoothing improves the classification accuracy of *every base kernel*.

In our second experiment, we picked the best smoothed kernel, in terms of classification accuracy, for each of our datasets from Table 1, and compared their performance with the state-of-the-art graph kernels (see Table 1). As can be seen, the smoothed kernels outperform other methods on all datasets, and the results are statistically significant on every dataset except PTC.

In our third experiment, we compared Pitman-Yor smoothed graphlet kernels to base graphlet kernel. As can be seen from Table 1, Pitman-Yor smoothed graphlet kernels are able to improve the performance of all datasets while achieving statistically significant improvements over a majority of them. However, it can also be seen that Pitman-Yor smoothed graphlet kernels are outperformed by Smoothed graph kernels introduced in Section 3.

Finally, as our fourth experiment, we test the performance of graph kernels when *edge* and *label* noise is presence where we randomly removed and added $\{10\%, 20\%, 30\%\}$ of the edges in each graph for edge-noise and randomly flipped $\{25\%, 50\%, 75\%\}$ of the node labels in each graph for label-noise by respecting the label-distribution. Figure 4 shows the performance of smoothed graph kernels under noise. As can be seen from the figure, smoothed graph kernels are able to significantly outperform their base variants.

# 8   Conclusion and Future Work

We presented a novel framework for smoothing graph kernels inspired by smoothing techniques from natural language processing and applied our method to state-of-the-art graph kernels. Our framework is rather general, and lends itself to many extensions. For instance, by defining domain-specific parent-child relationships, one can construct different DAGs with different weighting schemes for smoothing. Another interesting extension of our smoothing framework would be to apply it to graphs with continuous labels. While we restricted ourselves to graph kernels in this paper, our framework is applicable to any R-convolution kernel that uses a frequency-vector based representation.

# 9   Acknowledgments

We thank to anonymous NIPS reviewers for their constructive comments. We also thank to Hyokun Yun for his help on Pitman-Yor Processes, and Jiasen Yang for mathematical proofs. This work is supported by the National Science Foundation under grant No. #1219015.

## Footnotes

[1]See Table 2 in [3] for summarization of various smoothing algorithms using this general framework.

[2]We used Nauty [12] to obtain canonically-labeled isomorphic representations of graphlets.

[3]Implementations of original and smoothed versions of the kernels are publicly available at `http://web.ics.purdue.edu/~ypinar/nips`.