[Reviews · NeurIPS 2015]

Submitted by Assigned_Reviewer_1

This paper proposes graph kernels that take smoothed matching into account when computing a structural similarity, i.e., generated by graphlet kernels, Weisfeiler-Lehman subtree kernels, and shortest-path kernels.

Moreover, the paper proposes an extension of the Pitman-Yor process for smoothing subgraph distributions, which originates novel graph kernels.

The empirical evaluation shows that the proposed kernels outperform their non smoothed counterpart. Additionally, they also outperform other graph kernels and are competitive in terms of runtime.

The idea of using NLP smoothing for graph matching is interesting and reasonable. However, the model proposed by the authors is a subcase of a not-so-recent smoothed tree kernels. For example, see:

Danilo Croce, Alessandro Moschitti, and Roberto Basili. Structured lexical similarity via convolution kernels on dependency trees. In Proceedings of the 2011 Conference on Empirical Methods in Natural Language Processing, pages 1034-1046, Edinburgh, Scotland, UK., July 2011. Association for Computational Linguistics.

which proposed a smoothing function in the partial tree kernel by Moschitti, ECML 2006. Such function is more general than the smoothing function proposed in the submitted paper since it can be any valid kernel similarity.

Note that the smoothed partial tree kernel can be straightforwardly applied to DAG, see:

Aliaksei Severyn, Alessandro Moschitti: Fast Support Vector Machines for Convolution Tree Kernels. Data Mining Knowledge Discovery 25: 325-357, 2012.

Finally, there has been older work using tree kernels with similarities:

Stephan Bloehdorn and Alessandro Moschitti, Exploiting Structure and Semantics for Expressive Text Kernels, Proceeding of the Conference on Information Knowledge and Management, Lisbon, Portugal, 2007
Summary: Interesting idea but it seems a subcase of already proposed work. The paper would greatly improve if a comparison and discussion of the missing related work were added.

Submitted by Assigned_Reviewer_2

The paper presents a framework, applicable to three families of graph kernels, where smoothing techniques are introduced to create smoothed variants of graph kernels.

The paper is well-organized and easy to follow, however the main results are limited to small accuracy improvements.

+ I would like to see (less vague) statements on how the drawbacks of R-convolution based kernels (cited in the Introduction) are being addressed by the smoothing framework proposed in the paper;

+ Unfortunately, Table 1 shows very little improvement on most datasets, with the exception of WL + MUTAG?

Any comments on why this happens and/or why not on the other cases?

+ Diagonal dominance, power-law behavior, and partial similarities are more likely to occur when graphs contain more diverse/complex information and structure.

For example, when they are labeled with continuous labels.

Details/statistics about the dataset should not have been omitted;

+ Figures 1, 2, 3 are not very informative.

In fact figures 1 and 2 could be combined into a more clear and detailed example with lower k.
Summary: This paper presents an incremental work on graph kernels.

Despite being a clever idea, smoothed variants of 3 graph kernels on different datasets do not show significant improvement (Table 1).

Submitted by Assigned_Reviewer_3

The authors present a new framework for smoothing R-convolution kernels. They apply this idea to smooth various graph kernels. They present extensive results that demonstrate that smoothed kernels outperform their un-smoothed counterparts in term of accuracy. Finally they compare the new kernels in the presence of edge and label noise.

Pro: 1) The paper is well written, easy to read and the supplementary material is substantial. 2) The empirical results of Table 1 and Table 2 are convincing. 3) I applaud the authors for providing their source code (after acceptance).

Cons: 4) A complexity analysis and a minimum of the run time comparison should be part of the main paper. 5) I did not find the results in the presence of noise very convincing. 5.1) It seems that for many of these datasets, edges are undirected. For example, the protein dataset. For that reason, it makes no sense to flip them. For edge noise, the accuracy difference is almost always the same at 0% noise than at 30% noise. I would suggest trying to remove or to add edges instead of changing their directions. 5.2) I think results when vertices are randomly added or removed would be interesting.

Other comments / minor things: 6) I let to the authors to decide but combining Figure 1 and Figure 2 would make a stunning figure. Also, the last layer of the tree in Figure 2 could be removed, the edges are not visible. 7) Overall I feel the paper could be compressed a little to make place for some of the supplementary material. 8) The long sentence between line 233 and 237 is hard to read (too long). 9) Missing coma at line 267 10) The authors should clarify in the caption of Table 2 that "Smoothed" refers to the best kernel of Table 1. 11) Table 1 clearly does not fit the page.
Summary: Well written paper, with good empirical results. A few things to address but nothing that should prevent its acceptance.

Author Feedback
Author rebuttal: * Reviewer1

> However, the model proposed ..
This work is similar to ours in the sense that both methods
use the notion of smoothing for structured objects. The related
work section will be rewritten to take this work into account.
However, we believe that our method is fundamentally different
than theirs as follows: they solve the following problem: given
two trees, how similar are they? In contrast, we ask: given two
graphs, can we count sub-graphs and smooth the counts? The smoothing
is achieved by either using a pre-computed tree (graphlet kernels) or
by using a data dependent tree (WL and SP kernels). Once the counts
are smoothed, we discard the trees and compute the kernel via a dot
product of the smoothed counts.

* Reviewer2

> I would like to ...
We will make this explicit in the final version. Here are some brief
comments:

1) Dependency between sub-structures: Our framework considers
dependencies between sub-structures by explicitly encoding them into a DAG. These
dependencies are ignored by many existing R-convolution kernels.

2) Diagonal dominance: A subgraph is not only similar to itself but
also to its neighbors in the DAG. These "diffused" partial
matchings address diagonal dominance.

3) Power-law behavior: We take away probability mass from
sub-structures that occur frequently but are not informative (top
levels of the DAG) and redistribute it into informative
sub-structures (bottom levels). This prevents small and
uninformative subgraphs from dominating the kernel.

> Unfortunately, Table 1 ...
Many of the benchmark datasets are a decade old, and very well
studied. Therefore one cannot expect dramatic improvements. Also MUTAG
is a small dataset comparing to others (e.g. 188 graphs vs. 4110),
thus the relative improvement can't expected to be similar.
However, note that almost all of our improvements are statistically
significant at p < 0.05 (shown in bold in Table 1).

> Diagonal dominance, ...
This is definitely on our agenda for future work. The key challenge
here is to design a smoothing function that can respect continuous
labels.

* Reviewer3

> A complexity analysis ...
We will move this into the main body.

> It seems that ...
All our graphs have undirected edges. By 'flipping' we meant randomly
deleting and adding an edge. We will clarify this.

> I think results ...
If space permits, we will include these results.

* Reviewer4

> This reviewer believes ...
Unless we misunderstood, the reviewer is suggesting adding a
pseudo-count for smoothing (aka Laplace or more generally Dirichlet
smoothing). What we describe are much more sophisticated variants
of these simple ideas.

* Reviewer5

> I have to ...
In the original SP kernel, the contribution due to the pair (ABCD,
ABCE) = 0. In our case, it is non-zero. This is over and above the
contributions of the sub-paths, as noted by the reviewer. Longer
paths have more discriminative power, but may not match exactly. By
taking the into account, we can improve the similarity computation.

> Please comment on ...
See our response to Reviewer 1.

> It would be nice ...
Our work is similar to propagation kernels (PK) in the sense that both
works are designed to improve performance upon information loss (e.g.
noise vs. partially labeled graphs). They define graph features that
are counts of similar node-label distributions on the respective graphs
by using LSH. Our framework not only considers node label distributions,
but also explicitly incorporates structural similarity via the DAG.
Moreover, it seems like PK is only applied to WL kernel, whereas our work
is applied to 3 major graph kernel families.

> Your DAG ...
Both kernels exploit the same property: all subpaths of a
shortest-path (SP) are also SPs. The aim of the GraphHopper
work is to cache certain SP computations to speed up
kernel computation. In our case, we use this property to
smooth the counts.

Moreover, notice that their SP kernel do not compare paths in terms
of endpoints and lenghts, but compare nodes encountered while
'hopping' along SPs. Therefore, our DAG is different than ours since
we build the DAG on SP sub-structures (e.g. concatenated endpoints
and length of the path).

* Reviewer6

> However, it is ...
Please see Appendix 9.4 Table 5 under the columns labeled GK, SP, WL
DAG.

DAG generation costs:
1. GK. One time effort of O(2^k). Empirically ~6 seconds for
k=6.

2. SP. Linear processing after the SPs have been computed.

3. WL. See Appendix, Figure 5 (bottom figures).

> Also in some ...
We conjecture that this is primarily due to the differences in the
noise level present in different datasets.